# Association between Gut Microbiota and Emotional-Behavioral Symptoms in Children with Attention-Deficit/Hyperactivity Disorder

**DOI:** 10.3390/jpm12101634

**Published:** 2022-10-02

**Authors:** Min-Jing Lee, Hsin-Chih Lai, Yu-Lun Kuo, Vincent Chin-Hung Chen

**Affiliations:** 1Department of Psychiatry, Chang Gung Memorial Hospital, Chiayi Branch, Chiayi 613, Taiwan; 2School of Medicine, Chang Gung University, Taoyuan 333, Taiwan; 3Department of Medical Biotechnology and Laboratory Science, College of Medicine, Chang Gung University, Taoyuan 333, Taiwan; 4Microbiota Research Center and Emerging Viral Infections Research Center, Chang Gung University, Taoyuan 333, Taiwan; 5Central Research Laboratory, Xiamen Chang Gung Hospital, Xiamen 361, China; 6Department of Laboratory Medicine, Linkou Chang Gung Memorial Hospital, Taoyuan 333, Taiwan; 7Research Center for Chinese Herbal Medicine and Research Center for Food and Cosmetic Safety, College of Human Ecology, Chang Gung University of Science and Technology, Taoyuan 333, Taiwan; 8Biotools Co., Ltd., New Taipei City 221, Taiwan

**Keywords:** attention-deficit/hyperactivity disorder, gut microbiota, emotion, behavior

## Abstract

Previous studies have explored the role of the microbiome in attention-deficit/hyperactivity disorder (ADHD). However, whether the microbiome is correlated with emotional–behavioral disturbances, the most common comorbid symptom of ADHD, remains unclear. We established a cross-sectional study in which 6- to 18-year-old children with ADHD who were receiving no medication and a healthy control group of children without ADHD were recruited to analyze their microbiome composition. Microbiota of fecal samples were collected and analyzed using a 16s rRNA gene sequencing approach. In comparison with the healthy control group, the gut microbiota in children with ADHD exhibited significantly lower beta diversity. The abundance of the phylum Proteobacteria and the genera *Agathobacter*, *Phascolarctobacterium*, *Prevotella_2*, *Acidaminococcus*, *Roseburia*, and *Ruminococcus gnavus* group was increased in the ADHD group compared with the healthy group. Linear discriminant effect size (LEfSe) analysis was used to highlight specific bacteria phylotypes that were differentially altered between the ADHD and control groups. A regression analysis was performed to investigate the association between microbiota and emotional–behavioral symptoms in children with ADHD. A significant association was noted between withdrawal and depression symptoms and *Agathobacter* (*p* = 0.044), and between rule-breaking behavior and the *Ruminococcus gnavus* group (*p* = 0.046) after adjusting for sex, age, and the ADHD core symptoms score. This study advances the knowledge of how gut microbiota composition may contribute to emotional–behavioral symptoms in children with ADHD. The detailed mechanisms underlying the role of the gut microbiota in ADHD pathophysiology still require further investigation.

## 1. Introduction

Attention-deficit/hyperactivity disorder (ADHD) is the most common neurodevelopmental disorder among children and adolescents, with an estimated prevalence rate of approximately 3% to 10% globally [1]. The disorder is characterized by inattention, hyperactivity, difficulties in impulse control, and impaired academic achievement, social function, and emotional management [2,3]. In addition to these core symptoms, emotional dysregulation and poor behavior control are also common symptoms in ADHD. Such emotional–behavioral disturbances, although not reaching the diagnostic criteria of psychiatric disorders, still strongly affect patients with the disorder [4]. Comorbid emotional–behavioral problems in ADHD have been correlated with poor social adjustment, less frequent marriage, unemployment, poor peer relations, and poor schooling performance [5].

ADHD is a multiple pathogenetic disorder influenced by several genes and environmental factors [6,7]. In addition, recent investigations on the linkage between the gut microbiota and brain function have suggested that the gut microbiota may play a role in the etiology of ADHD [8]. The bidirectional communication between the gut and the brain, also known as the “gut–brain axis”, has been proposed to be involved in some neuropsychiatric disorders, including depression, anxiety, and schizophrenia [9,10,11]. Furthermore, the gut microbiota has also been reported to affect host development and physiology, which is linked to autism and ADHD [12,13]. The evidence indicates that the gut microbiota may release neuromodulators, including dopamine and serotonin, which affect emotional regulation [14,15]. However, how the gut microbiota affects emotional–behavioral symptoms in patients with ADHD still requires clarification.

Only a few studies have investigated the gut microbial profiles of patients with ADHD, with the results being inconsistent. Some research has compared patients with ADHD and healthy controls but without demonstrating significant differences in alpha and beta diversity between the two cohorts [16,17,18]. However, one study that analyzed the microbial composition of a German population of adolescents with ADHD and nonrelated controls reported the conflicting result. Alpha and beta diversity were both significantly different between the two cohorts, and alpha diversity was negatively correlated with the level of hyperactivity [19]. Another study in Taiwan revealed no differences in beta diversity between children with ADHD and healthy controls. However, ADHD children had significantly higher values of Shannon and Chao 1 indices, and the Simpson index was significantly lower compared with controls [2]. Therefore, it is difficult to draw a conclusion from these studies due to the heterogenous composition of the study populations. 

Studies have examined the correlation between the gut microbiota and clinical symptoms among patients with ADHD [2,17,19,20]. However, in addition to the core symptoms of ADHD, the association between the gut microbiota and emotional–behavioral problems of patients with ADHD also require an investigation. High comorbid mood and behavior disorder has been reported but the role of the gut microbiota in emotional–behavioral symptoms among patients with ADHD has not been explored [21]. We thus performed a study to determine whether an imbalanced gut microbiota would be correlated with the emotional–behavioral symptoms in patients with ADHD. We compared the gut microbiota composition of healthy children with that of patients with ADHD and examined the correlation between the gut microbiota composition and emotional–behavioral symptoms. 

## 2. Materials and Methods

### 2.1. Participants

All participants were recruited at Chiayi Chang Gung Memorial Hospital. The participants were aged 6-to-18 years old, and participants and their parents agreed to provide written informed consent. Patients with ADHD were diagnosed according to the DSM-5 criteria by a senior psychiatrist [22]. All participants were assessed using the Swanson, Nolan and Pelham Parent Rating Scale. The exclusion criteria included taking probiotics in the last 2 weeks and having gastrointestinal disease or other developmental or psychiatric disorders (except oppositional defiant disorder and conduct disorder). To avoid the effect of medication, participants who received ADHD medication were also ruled out. General patient information, including sex, age, body mass index (BMI), and a history of preterm birth or rhinitis, was collected in this study. The Institutional Review Board (IRB) at Chang Gung Memorial Hospital in Taiwan approved our research protocol (IRB number: 201800402A3).

### 2.2. Sample Collection and DNA Extraction

To guarantee the reliability of the data, quality control was performed at each step of study process (ie, DNA sampling, PCR testing, library preparation and sequencing). Total genomic DNA from samples was extracted using the column-based method (eg, QIAamp PowerFecal DNA Kit, Qiagen, Hilden, Germany). DNA concentration was determined and adjusted to 5 ng/μL for the sequencing process.

### 2.3. Sequencing

For the 16S rRNA gene sequencing, the V3-V4 region was amplified by a specific primer set (314F: TCGTCGGCAGCGTCAGATGTGTATAAGAGACAGCCTACGGGNGGCWGCAG, 805R: GTCTCGTGGGCTCGGAGATGTGTATAAGAGACAGGACTACHVGGGTATCTAATCC) according to the 16S Metagenomic Sequencing Library Preparation procedure (Illumina, San Diego, CA, USA). In brief, 5 ng of gDNA was used for the PCR reaction, which was conducted using KAPA HiFi HotStart ReadyMix (Roche, Basel, Switzerland) under the following PCR conditions: 95 °C for 3 min; 25 cycles of 95 °C for 30 s, 55 °C for 30 s, and 72 °C for 30 s; 72 °C for 5 min; and hold at 4 °C. The PCR products were monitored on 1.5% agarose gel. Samples with a bright main strip at 550 bp were chosen and purified by using AMPure XP beads for the following library preparation.

The sequencing library was prepared according to the 16S Metagenomic Sequencing Library Preparation procedure (Illumina). In brief, a secondary PCR was performed using the 16S rRNA V3-V4 region PCR amplicon and Nextera DNA Indexes Kit with dual indices and Illumina sequencing adapters (Illumina). The indexed PCR product quality was assessed using the Qubit dsDNA High Sensitivity Quantification Assay (Thermo Scientific, Waltham, MA, USA) and Qubit Fluorometric 2.0. An equal amount of the indexed PCR product was mixed to generate the sequencing library. Lastly, the library was sequenced on an Illumina MiSeq platform, and paired 300-bp reads were generated.

### 2.4. Data Processing and Analysis

Amplicon sequencing was performed by using 300-bp paired-end raw reads, and the entire paired-end reads were assembled using FLASH v1.2.11 [23]. Demultiplexing was performed using barcode identification. As part of quality control, low-quality reads (Q < 20) were discarded in the QIIME v1.9.1 pipeline [24]. If 3 consecutive bases were Q < 20, the read was truncated and the resulting read remained in the data set only if it was at least 75% of the original length, which was determined using the split_libraries_fastq.py script in QIIME. Sequences were chimera-checked using UCHIME [25] to obtain the effective tags and were filtered from the data set before operational taxonomic unit (OTU) clustering at 97% sequence identity by using the UPARSE function in the pipeline. For each representative sequence, the RDP classifier (v2.2) [26] algorithm was employed to annotate the taxonomy classification on the basis of the information retrieved from the Silva Database v132: 2017.12 [27]/GreenGenes vgg_13_8 [28,29]/NCBI/eHOMD v15.1 [30,31]; classification was performed, and an 80% minimum confidence threshold was necessary to make an assignment. Sequences with a one-time occurrence (singletons) or that were present in only one sample were filtered out. To analyze the sequence similarities among different OTUs, a multiple sequence alignment was conducted using the PyNAST software (v1.2) [32,33]; sequences were compared against the core-set dataset in the Silva Database v132; 2017.12/GreenGenes vgg_13_8/NCBI/eHOMD v15.1. A phylogenetic tree was constructed with a set of sequences representative of the OTUs by using FastTree [34,35].

### 2.5. Measurements

The severity of ADHD symptoms, including inattention, hyperactivity, and impulsivity, was measured and rated by parents with the SNAP-IV parent form [22,36]. The emotional–behavioral symptoms were also measure by parents using the Child Behavior Checklist (CBCL). The CBCL is a practical instrument used by parents to evaluate the emotional and behavioral symptoms of children between 4 and 16 years old. The CBCL contains 8 narrowband syndromes (i.e., withdrawn, anxious/depressed, somatic complaints, social problems, thought problems, attention problems, aggressive behavior, and delinquency) and 2 broadband syndromes (i.e., internalizing problems and externalizing problems). A T-score of 50 in each subscale indicates average functioning in reference to other children of the same age and gender. The Chinese language version of the CBCL possesses high test-retest reliability and validity [37,38].

### 2.6. Statistical and Bioinformatic Analysis

Microbiome 16S sequencing data were analyzed using a combination of the software programs mothur and R v3.2.5 [39]. To compare the microbial differences between groups, alpha diversity and beta diversity were analyzed. Alpha diversity, which represents the complexity of composition within members of a group, was calculated using the metrics of Chao 1 and ACE (community richness) and the Shannon and Simpson indices (the relative abundance and evenness accounting for diversity). Beta diversity, which represents the between-subject similarity of a microbial composition and enables the identification of differences between samples within a group, was estimated using UniFrac (Unique Fraction) [40,41]. We used an unweighted UniFrac distance that considered only information on the presence and absence of a species and counted the fraction of branch length unique to either community and a weighted UniFrac distance that used species abundance information and weighted the branch length with the difference in abundance. To determine whether the UniFrac distances were on average significantly different for pairs of samples, a *t* test was performed on the UniFrac distance matrix. To further increase the group distinction, supervised partial-least-squares discriminant analysis (PLS-DA) was used to evaluate and visualize variance between the groups on the basis of gut microbiota composition (OTUs). PLS-DA, which is based on a least-squares regression model, indicated structural differences in the gut bacterial community between the groups. These differences were assessed by calculating UniFrac distances, a phylogenetic-based distance metric for measuring the difference between two collections of sequences.

The composition analysis was calculated using QIIME V1.9.1. We used a linear discriminant analysis effect size (LEfSe) for statistical analysis and visualizing the results [42]. LEfSe takes into account both differences in abundance and frequency and was used to identify significantly imbalanced OTUs between the ADHD and healthy groups. 

Clinical data were analyzed using the statistical software package SPSS, version 20 (SPSS, Chicago, IL, USA). Data were presented as mean ± SD, median (interquartile range), or number (percentage), as appropriate. Basic characteristics of the children with ADHD and the healthy controls were compared using chi-square tests for categorical variables, independent Student’s *t* tests for normally distributed continuous variables, and Mann–Whitney U tests for skewed continuous variables. Pearson’s correlation analysis was performed to assess the taxonomic association between gut microbiota and clinical symptoms. Multiple linear regression was performed to analyze the relationship between the relative abundance of bacteria and the CBCL rating score after controlling for sex, age, and SNAP-IVscore. 

## 3. Results

### 3.1. Demographic Data

A total 371 participants were recruited, and 270 participants were excluded due to taking probiotics, having gastrointestinal disease, developmental or psychiatric disorders. There were 79 children with ADHD enrolled and 25 children ADHD who received medication among them were excluded. Finally, a total of 54 children with ADHD (boys = 44, girls = 10) without drug treatment and 22 healthy controls (boys = 13, girls = 9) were enrolled (Figure 1). The proportion of male was significant higher in ADHD cohort than that of the healthy control group (*p* = 0.042). The mean age of the ADHD cohort (mean = 8.39 ± 1.75) was younger than that of the healthy control group (mean = 9.73 ± 2.23). No significant differences were apparent between the groups in terms of BMI, or a history of preterm birth or allergic rhinitis. We compared the core ADHD symptoms through SNAP-IV ratings assigned by parents. Scores for inattention, hyperactivity/impulsivity, ADHD total, opposition/defiant, and CBCL were higher in children with ADHD compared with the healthy controls. The characteristics of the children with ADHD and those of the healthy controls are summarized in Table 1.

### 3.2. Alpha Diversity and Beta Diversity

Microbial diversity was assessed either within a community (alpha diversity) or between the collection of samples (beta diversity). Alpha diversity indices, including the Simpson index, the Shannon diversity index, the ACE metric (Abundance-based Coverage Estimator), and Chao 1, were calculated using Kruskal-Wallis LSD post-hoc tests. No significant difference was evident between the children with and without ADHD in alpha diversity. Subsequently, we characterized beta diversity using UniFrac and PLS-DA and determined that the gut microbiota communities of the children with ADHD and the healthy controls were significantly different (unweighted UniFrac, *p* = 0.00; weighted UniFrac, *p* = 0.00; Figure 2 and Figure 3). Children without ADHD had larger variation in their gut microbiota composition than the ADHD children did.

### 3.3. LEfSe Analysis

We used LEfSe analysis to identify the specific bacteria phylotypes that were differentially altered between the ADHD and control groups (Figure 4). Significant differences were evident in the microbial community compositions of the two groups (the logarithmic LDA score threshold for discriminative features was set to 3.0). At the genus level, the relative abundance of Alistipes and the Eubacterium eligens group in the healthy group was significantly higher than in the ADHD group. At the family level, the relative abundance of Rikenellaceae in the healthy group was significantly higher than in the ADHD group. At the genus level, Acidaminococcaceae, Agathobacter, Phascolarctobacterium, Prevotella 2, Parasutterella, Acidaminococcus, Roseburia, and the Ruminococcus gnavus group were more abundant in the ADHD group than in the healthy group. Class Gammaproteobacteria, order Betaproteobacteriales, family Burkholderiaceae, and genus Parasutterella all belong to the phylum Proteobacteria (Figure 4).

### 3.4. Association between Clinical Symptoms and Relative Abundance of Bacteria

Table 2 shows the correlation between the relative abundance of bacteria and ADHD symptoms among participants with ADHD. No significant relationship was identified between SNAP-IV score and each bacteria taxon. 

Table 3 presents the correlations between the relative abundance of bacteria and the CBCL scores of participants with ADHD. Withdrawal and depression symptoms were associated with Agathobacter (*p* = 0.02) and Acidaminococcus (*p* = 0.035). Somatic complaints were associated with Rikenellaceae (*p* = 0.016) and Alistipes (*p* = 0.016). Thought problems were associated with Acidaminococcus (*p* = 0.021). In addition to rule-breaking behavior, aggressive behavior and externalizing behavior were associated with the Ruminococcus gnavus group (*p* = 0.046, 0.031, and 0.036, respectively). 

Furthermore, we examined the effect of each bacteria taxon on the CBCL symptoms among children with ADHD. We chose the bacteria which is more abundant among ADHD cohort than that of healthy control, including Agathobacter, Acidaminococcus and Ruminococcus gnavus group, and examined the association between the selected bacteria taxon and their correlated CBCL score by using a regression analysis and adjusting for sex, age, and SNAP-IV scores (Table 4). A significant association was noted between withdrawal and depression symptoms and Agathobacter (*p* = 0.044), and between rule-breaking behavior and the Ruminococcus gnavus group (*p* = 0.046). 

## 4. Adjusted for Sex, Age and Score of SNAP-Iv Discussion

To the best of our knowledge, this is the first study to investigate the association between the selected bacteria and emotional–behavioral symptoms in children with ADHD. In the present study, we determined that Agathobacter, Acidaminococcus, Rikenellaceae, and Alistipes were associated with internalizing symptoms and the Ruminococcus gnavus group was associated with externalizing symptoms in children with ADHD. After controlling for confounding factors, including basic characteristics and SNAP-IV scores, the results indicated that Agathobacter was associated with withdrawal and depression symptoms and the Ruminococcus gnavus group was associated with rule-breaking behavior. 

Previous research has reported that several bacteria were associated with clinical ADHD symptoms [2,19,20]. One Taiwanese study revealed that the relative abundance of Bacteroides was associated with clinical ADHD symptoms rated by SNAP-IV [2]. Another study from Germany also determined that Bacteroides was correlated with relatively high levels of hyperactivity and impulsivity [19]. However, they reported no significant correlations between the microbiome and the CBCL symptoms. A study in the Netherlands revealed that Ruminococcus was associated with inattention [20]. In our study, we did not identify any correlation between the bacteria and core symptoms of ADHD but did determine that Agathobacter and the Ruminococcus gnavus group were associated with CBCL symptoms. The discrepancy between our study and the Western studies may be attributed to ethnic differences and diet. The discrepancy between our study and another Taiwanese study may need further investigation with a larger sample size.

In our study, we found that the Ruminococcus gnavus group was associated with externalizing behavior (per the CBCL) in children with ADHD. Ruminococcus gnavus, an anaerobic, gram-positive microbiome, is common in the human gut [43]. Ruminococcus gnavus has been associated with several inflammatory diseases, including spondyloarthritis, eczema in infants, and Crohn disease [44,45,46,47,48,49]. Ruminococcus gnavus belongs to the family Ruminococcacceae, and the relative abundance of Ruminococcacceae has been reported to be linked to several psychiatric diseases, such as mood disorders, schizophrenia, autism, and ADHD [15,50,51,52]. One animal study reported that mice colonized with ADHD gut microbiota experienced an increase in anxious behavior and their structural and functional brain characteristics were affected [53]. Ruminococcus is involved in the metabolic pathway of complex sugar degradation and the degradation of mucin [54,55], which is vital for supplementing energy. Short-chain fatty acids (SCFAs), which are metabolites of Ruminococcus, are known to be key molecules that affect brain function and human behavior and potentially play a role in ADHD [56,57]. Further studies exploring how Ruminococcus affects emotional–behavioral symptoms in children with ADHD are required. Our study also noted that Agathobacter was associated with withdrawal and depression symptoms in children with ADHD. Agathobacter is an anaerobic, gram-positive bacterial genus from the family Lachnospiraceae. Similar to Ruminococcus, Agathobacter has been found to produce SCFAs [58,59]. Research has suggested that the level of SCFAs was relevant to inflammatory bowel conditions with comorbid depression and anxiety [60,61]. The abundance of Agathobacter has also been determined to be associated with ADHD and sleep problems in a population with autism spectrum disorder [62,63]. There is a paucity of information on the role of Agathobacter in the pathophysiology of depression. The mechanism underlying the association between Agathobacter and withdrawal and depression symptoms in children with ADHD requires further exploration. 

### Strengths and Limitations

This study has several limitations that should be noted. First, this study was conducted using a cross-sectional method and was unable to determine longitudinal changes and factors associated with the gut microbiota. Second, although we observed that several bacterial genera were correlated with ADHD clinical symptoms, no specific bacteria species were identified. Further gene mapping of bacteria is necessary to explore the mechanisms underpinning the association between the gut microbiota and ADHD. Third, we did not investigate diet in this study. Some specific diets (ie, vegan, gluten-free, paleo, low-carb, isoflavone intake) may influence the microbiota. Previous studies have reported that the Oriental diet is abundant in isoflavone, which may interact with microbiota and produce bioactive compounds contributing benefits including inducing nitric oxide production and anti-inflammatory effect on human health [64,65]. More research is needed to investigate the antioxidant capacity and anti-inflammatory effect of microbiota and elaborate the possible para-probiotics compounds to ameliorate emotional–behavioral symptoms in children with ADHD. 

## 5. Conclusions

In conclusion, we determined that gut microbiome variability differs between children with ADHD and a healthy population. Two specific genera, Ruminococcu and Agathobacter, were determined to be associated with the emotional–behavioral symptoms of children with ADHD. Further study to explore the gut microbiome on ADHD should considered the effect of comorbid emotional–behavioral symptoms. The specific mechanisms responsible for the association between the gut microbiota and ADHD still require further investigation.

## Figures and Tables

**Figure 1 jpm-12-01634-f001:**
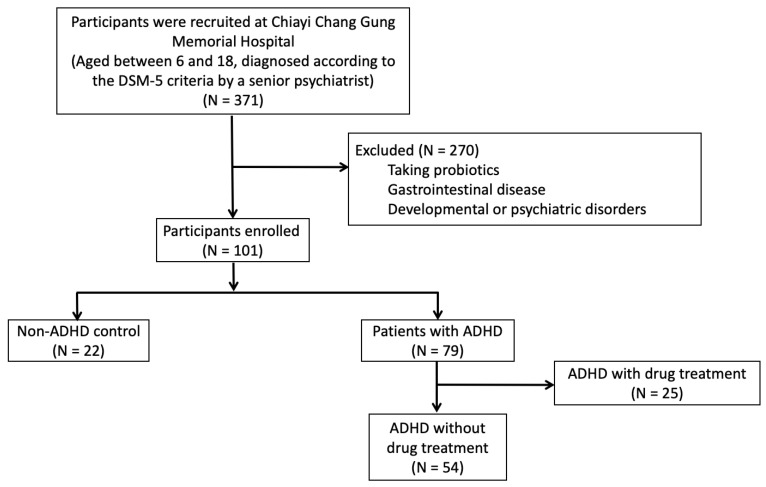
Flowchart of enrollment of the study participants.

**Figure 2 jpm-12-01634-f002:**
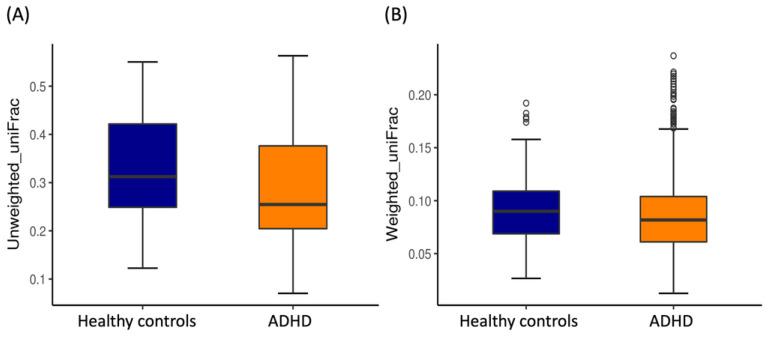
Beta diversity using unweighted and weighted uniFrac. Comparison of beta diversity measurements in ADHD children with treatment-naïve and healthy controls. Beta diversity, measured by unweighted (**A**) and weighted (**B**) unique fraction (UniFrac) is plotted for ADHD children with treatment-naïve (orange) and healthy controls (blue). The line inside the box represents the median, while the whiskers represent the lowest and highest values within the 1.5 interquartile range. Outliers as well as individual samples are shown as dots. The results indicated that there was significant difference in the gut microbiota communities between ADHD children with treatment-naïve and healthy controls.

**Figure 3 jpm-12-01634-f003:**
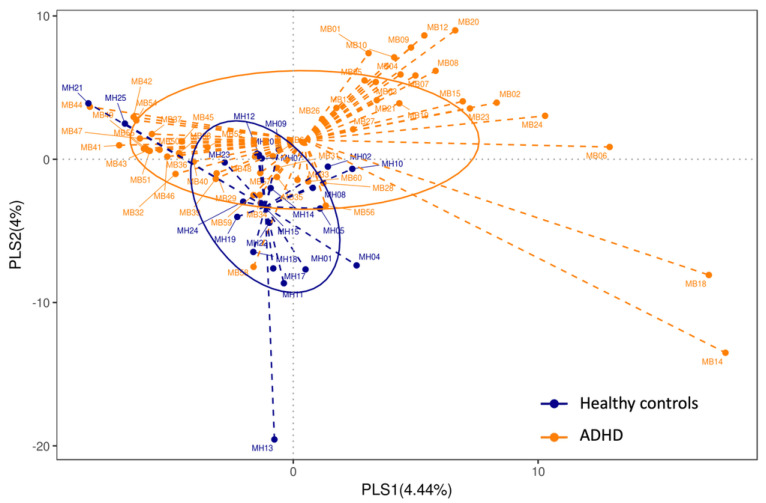
PLS−DA. Partial Least Squares Discriminant Analysis (PLS−DA) of ADHD children with treatment-naïve and healthy controls. Each point represents a sample. The orange points represent ADHD children with treatment-naïve and the blue points represent healthy controls.

**Figure 4 jpm-12-01634-f004:**
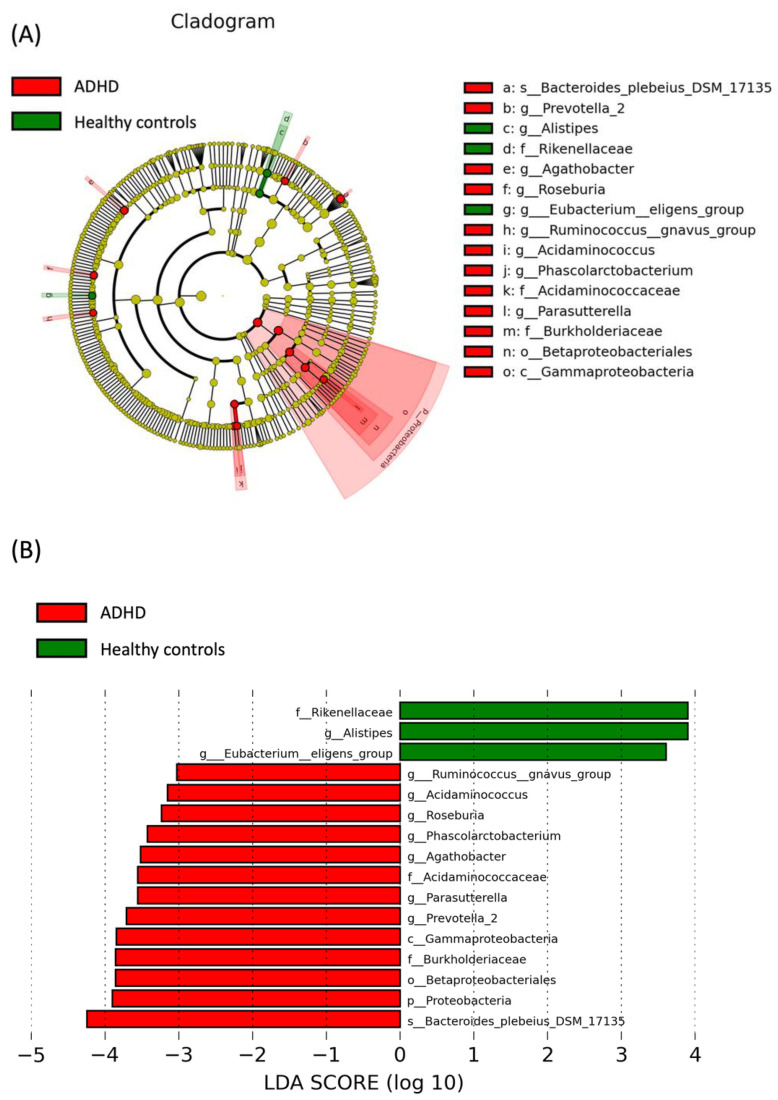
Linear discriminant effect size (LEfSe) analysis. Comparison of microbiota relative abundance at the genus level between ADHD children with treatment-naïve and healthy controls using LEfSe cladograms (**A**) and histogram of linear discriminant analysis (LDA, **B**). The bacterial difference was identified by Kruskal-Wallis test. Nominal significant threshold: *p* < 0.05.

**Table 1 jpm-12-01634-t001:** Baseline characteristics of ADHD children with treatment-naïve and health controls.

	ADHD Children with Treatment-Naïve (N = 54)	Healthy Controls (N = 22)	*p*-Value	Effect Size ^a^
Sex, n (%)			0.042 *	3.046
Male	44 (81.5%)	13 (59.1%)		
Female	10 (18.5%)	9 (40.9%)		
Age (mean ± SD year)	8.39 (1.75)	9.73 (2.23)	0.007 *	0.669
BMI (mean ± SD)	18.32 (3.91)	19.87 (3.72)	0.112	0.406
Preterm birth, n (%)	5 (9.3%)	2 (9.1%)	0.675	1.020
Allergic rhinitis, n (%)	19 (35.2%)	7 (31.8%)	0.499	1.163
SNAP-IV				
Inattention (SD)	15.15 (5.7)	4.5 (4.53)	<0.001 *	2.069
Hyperactivity/Impulsivity (SD)	11.65 (6.37)	3.41 (4.67)	<0.001 *	1.475
Opposition/defiance (SD)	9.83 (6.26)	5.64 (5.28)	0.007 *	0.724
Total Score (SD)	26.8 (10.58)	7.91 (8.19)	<0.001 *	1.997
CBCL syndromes, median (SD)				
Anxious/Depressed	59.15 (8.72)	54.91 (7.09)	0.047 *	0.534
Withdrawn/Depressed	60.41 (9.26)	55.82 (5.35)	0.033 *	0.607
Somatic Complaints	56.7 (5.79)	53.95 (5.75)	0.064	0.477
Social Problems	63.72 (8.33)	55.5 (8.76)	<0.001 *	0.962
Thought Problems	59.52 (7.91)	53.45 (5.35)	0.002 *	0.899
Attention Problems	68.93 (9.8)	55.45 (6.06)	<0.001 *	1.654
Rule-Breaking Behavior	62.2 (7.94)	53.86 (5.33)	<0.001 *	1.233
Aggressive Behavior	62.59 (8.56)	53.91 (5.42)	<0.001 *	1.212
Internalising	58.48 (10.54)	51.36 (10.09)	0.009 *	0.690
Externalising	62.41 (9.36)	49.77 (9.05)	<0.001 *	1.373
Total Score	64.22 (8.63)	50.09 (11.21)	<0.001 *	1.413

* *p* < 0.05; ^a^ Effect size was analyzed using Cohen’s d for continuous variables and odds ratio for categorical variables.

**Table 2 jpm-12-01634-t002:** The correlation between relative abundance of microbiota and clinical symptoms of attention-deficit/hyperactivity disorder (ADHD) according to SNAP-IV.

Genera	Inattention		Hyperactivity/Impulsivity		Total Score	
	*r*	*p*	*r*	*p*	*r*	*p*
*Proteobacteria*	0.018	0.899	0.012	0.932	0.041	0.768
*Gammaproteobacteria*	−0.065	0.641	−0.030	0.829	−0.022	0.874
*Betaproteobacteriales*	0.078	0.575	−0.082	0.558	0.007	0.959
*Burkholderiaceae*	0.078	0.575	−0.082	0.558	0.007	0.959
*Acidaminococcaceae*	−0.107	0.443	−0.125	0.368	−0.120	0.389
*Agathobacter*	0.236	0.086	0.176	0.204	0.236	0.086
*Phascolarctobacterium*	−0.070	0.617	−0.109	0.433	−0.097	0.485
*Prevotella_2*	0.134	0.336	−0.071	0.612	0.021	0.881
*Parasutterella*	0.108	0.437	−0.042	0.762	0.037	0.793
*Acidaminococcus*	0.116	0.402	0.082	0.558	0.124	0.372
*Roseburia*	0.024	0.864	0.134	0.333	0.090	0.518
*Ruminococcus_gnavus_group*	−0.054	0.700	0.070	0.615	0.016	0.906
*Bacteroides_plebeius_DSM_17135*	0.157	0.256	0.039	0.779	0.094	0.498
*Rikenellaceae*	−0.074	0.743	−0.022	0.921	−0.032	0.888
*Alistipes*	−0.074	0.743	−0.022	0.921	−0.032	0.888
*Eubacterium_eligens_group*	0.289	0.191	−0.057	0.801	0.124	0.582

**Table 3 jpm-12-01634-t003:** The correlation between relative abundance of microbiota and comorbid emotional-behavioral symptoms of attention-deficit/hyperactivity disorder (ADHD) according to Child Behavior Checklist (CBCL).

Genera	Withdrawn /Depressed		Somatic Complaints		Social Problems		Thought Problems		Attention Problems		Rule-Breaking Behavior		Aggressive Behavior		Internalizing		Externalizing		Total Score	
	*r*	*p*	*r*	*p*	*r*	*p*	*r*	*p*	*r*	*p*	*r*	*p*	*r*	*p*	*r*	*p*	*r*	*p*	*r*	*p*
*Proteobacteria*	0.210	0.127	0.225	0.102	0.153	0.268	0.119	0.393	0.152	0.273	0.106	0.446	0.162	0.242	0.222	0.107	0.148	0.284	0.192	0.165
*Gammaproteobacteria*	0.197	0.154	0.190	0.168	0.180	0.194	0.081	0.560	0.126	0.364	0.109	0.433	0.155	0.263	0.210	0.127	0.150	0.280	0.180	0.193
*Betaproteobacteriales*	0.057	0.683	0.023	0.870	−0.071	0.610	−0.053	0.705	−0.142	0.305	−0.043	0.757	−0.067	0.630	−0.025	0.860	−0.076	0.586	−0.095	0.495
*Burkholderiaceae*	0.057	0.683	0.023	0.870	−0.071	0.610	−0.053	0.705	−0.142	0.305	−0.043	0.757	−0.067	0.630	−0.025	0.860	−0.076	0.586	−0.095	0.495
*Acidaminococcaceae*	−0.039	0.779	0.073	0.602	−0.077	0.582	0.122	0.380	−0.026	0.853	−0.049	0.726	−0.051	0.716	0.040	0.774	−0.057	0.680	−0.030	0.832
*Agathobacter*	0.317	0.020 *	0.119	0.391	0.241	0.079	0.250	0.069	0.120	0.389	0.162	0.241	0.083	0.551	0.146	0.292	0.109	0.431	0.184	0.184
*Phascolarctobacterium*	−0.139	0.318	0.104	0.454	−0.161	0.244	0.042	0.762	−0.049	0.727	−0.132	0.341	−0.135	0.331	−0.032	0.816	−0.144	0.299	−0.107	0.442
*Prevotella_2*	0.000	0.999	−0.019	0.894	0.095	0.494	−0.019	0.890	−0.097	0.485	0.081	0.562	0.061	0.661	−0.038	0.788	0.097	0.486	0.029	0.838
*Parasutterella*	−0.044	0.750	0.006	0.963	−0.155	0.264	−0.017	0.901	−0.156	0.259	−0.160	0.248	−0.134	0.335	−0.060	0.668	−0.161	0.246	−0.148	0.287
*Acidaminococcus*	0.288	0.035 *	0.082	0.556	0.223	0.105	0.313	0.021 *	0.202	0.143	0.224	0.104	0.204	0.140	0.154	0.265	0.221	0.108	0.244	0.076
*Roseburia*	−0.058	0.678	−0.103	0.461	0.104	0.456	0.003	0.986	−0.073	0.599	0.058	0.678	0.030	0.828	−0.095	0.493	0.056	0.685	0.010	0.943
*Ruminococcus_gnavus_group*	0.195	0.157	0.151	0.277	0.175	0.205	0.264	0.054	0.242	0.077	0.272	0.046 *	0.295	0.031 *	0.251	0.068	0.285	0.036 *	0.291	0.033 *
*Bacteroides_plebeius_DSM_17135*	−0.033	0.814	0.027	0.845	−0.080	0.567	0.008	0.953	0.058	0.678	−0.104	0.454	−0.113	0.416	−0.060	0.664	−0.090	0.518	−0.035	0.799
*Rikenellaceae*	−0.131	0.562	0.505	0.016 *	0.119	0.598	0.088	0.697	0.058	0.799	−0.138	0.540	0.162	0.472	0.341	0.121	0.038	0.868	0.225	0.315
*Alistipes*	−0.131	0.562	0.505	0.016 *	0.119	0.598	0.088	0.697	0.058	0.799	−0.138	0.540	0.162	0.472	0.341	0.121	0.038	0.868	0.225	0.315
*Eubacterium_eligens_group*	−0.224	0.317	−0.112	0.620	0.048	0.832	0.108	0.634	−0.022	0.923	−0.112	0.620	−0.210	0.348	−0.113	0.615	−0.217	0.332	−0.100	0.658

* *p* < 0.05.

**Table 4 jpm-12-01634-t004:** Multiple regression analysis of the taxonomic association between gut microbiota and comorbid emotional-behavioral symptoms of attention-deficit/hyperactivity disorder (ADHD) according to Child Behavior Checklist (CBCL).

Genera	Withdrawn/Depressed			Thought Problems			Rule-Breaking Behavior			Aggressive Behavior		
	B	SE	*p* Value	B	SE	*p* Value	B	SE	*p* Value	B	SE	*p* Value
*Agathobacter*	0.002	0.001	0.044 *									
*Acidaminococcus*	0.003	0.002	0.124	0.002	0.001	0.079						
*Ruminococcus_gnavus_group*							0.005	0.003	0.046 *	0.004	0.003	0.111

* *p* < 0.05.

## Data Availability

The data that support the findings of this study are available on request from the corresponding author. The data are not publicly available due to privacy or ethical restrictions.

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
