# Peer review of "Association between Gut Microbiota and Emotional-Behavioral Symptoms in Children with Attention-Deficit/Hyperactivity Disorder"

_jpm, 2022, doi:10.3390/jpm12101634_

Round 1
Reviewer 1 Report
Comments to the Author
The present manuscript deals with an interesting topic concerning between gut microbiota and emotional-behavioral symptoms in children with ADHD. This study has several strengths, such as a relatively large and well identified ADHD sample, and sophisticated techniques to collect biological information. However, I think that some points should be clarified.
#1. ADHD medication
The sample is composed of 54 children with ADHD and none is taking medication. It is known that more than 50% of children with ADHD take some medication (e.g., methylphenidate). The ‘Participants’ section should be clearer whether this was an exclusion criteria and, if it was the case, how many children were discarded from the study due to this reason. Generally, a flow chart describing the recruitment process should be an interesting figure to include.
#2. CBCL.
In my experience, CBCL completed by parents without any assistance can sometimes have a low internal consistency. Can the authors report the Cronbach’s alpha for each of the 10 subscales? (8 narrow, 2 broad)
#3. Reproducibility.
The raw data (if possible), and the code used to perform analyses should be shared in Supplementary Materials.
#4. Demographic.
Authors have chosen to present a tableone to compare demographic characteristics of the samples. They rely on p-values to determine whether the samples differ. I think this is something quite tentative given the low number of controls. Authors should also present effect sizes to give a more interesting picture of the differences between the groups.
Moreover, the authors rely on p-values but consider that there were no gender differences (“No significant differences were apparent between the groups in terms of gender”) while there were significantly more male than female in the sample. I guess this is an error, but the authors should be clearer on whether the sex or gender was collected and compared.
#5. Objectives.
The main objective of the paper was : “Studies have examined the correlation between the gut microbiota and clinical symptoms among patients with ADHD [2,17,19,20]. However, in addition to the core symptoms of ADHD, the emotional–behavioral problems of patients with ADHD also require an investigation”
However, in the results, in the abstract, etc. the main result presented is the difference between ADHD and control children regarding hut microbiota composition. Authors should better hierarchize their objectives and provide a better justification of the case-control design used if their main objective was simply to observe an association in an ADHD sample.
#6. Multiplicity testing
As far I as understand, the core objective was to assess the associations between gut mcrobiota and some subscales of the CBCL in an ADHD sample. However, as can be shown in table 3, authors have tested dozens of associations (16*10 = 160 associations). It is therefore not surprising that they found some nominally significant associations. Authors should find a strategy to address this major issue (ideally, they should have selected some primary associations in a registered protocol but, as far as I understand, this was not performed) because one cannot conclude on 10 statistically significant associations while testing more than 150.
#7. Header table 3
The header of table 3 is incorrect or misleading (I think that clinical symptoms of attention-deficit/hyperactivity disorder (ADHD) should be renamed as in the rest of the paper or simply CBCL subscales, scores, etc.). The population (only I ADHD children) should be indicated.
#8. Table 4.
I did not manage to understand how the authors have proceeded to select the variables tested in the table 4. They should present their approach more clearly and specify whether they included each predictor one-by-one or if they ran some models with more than one independent variable (in addition to the control variables)
#9. Discussion
I would appreciate to re-read the discussion once the different issues raised are resolved (in particular regarding the multiplicity testing issue that prevent relying on the results as such)
Author Response
We thank the reviewer for the valuable comments. We have done our best to address all of the reviewers’ comments. Please see the attachment.

Reviewer 2 Report
In the present paper, Min-Jing Lee and colleagues performed a study to determine whether an imbalanced gut microbiota would be correlated with the emotional-behavioral symptoms in patients with attention-deficit/hyperactivity disorder (ADHD). Specifically, they compared the gut microbiota composition of healthy children with that of patients with ADHD and examined the correlation between the gut microbiota composition and emotional-behavioral symptoms. The authors concluded that the present findings advance the knowledge of how gut microbiota composition may contribute to emotional-behavioral symptoms in children with ADHD. However, the detailed mechanisms underlying the role of the gut microbiota in ADHD pathophysiology still require further investigation. Overall, I think that the paper could be of interest for readers and researchers, in general, on a current topic of interest.
I make some suggestions for improve the quality of the manuscript.
1) Gut microbioma composition is related to dietary habit. So, have you analyzed the nutritional status of patients included in this analysis? Please explain. The authors, if possible, should incorporate in tables the dietary pattern of the patients included in the present study (e. Mediterranean-style diet, Plants-based diet, Nordic dietary pattern, etc.); in this way, I feel that the readers can better understand the results obtained in the present clinical study and their possible application to clinical practice.
2) “Oriental diet” is particularly abundant in isoflavones. So, the intriguing effects on emotional-behavioral symptoms as well as on gut microbiota composition observed in the present study could be related to these phytochemicals. Please discuss this aspect in the manuscript considering for your convenience these references (D’Anna, R. et al. Menopause, 2009, 16, 301-306; Ma L. et al. Circulation. 2020, 141, 1127-1137; Marini, H.R. Nutrients, 2022, 14, 1550).
3) Please better define and discuss the power analysis of study.
4) In light of the results here obtained, please to discuss on the possible application of nutraceutics and/or antioxidants/antinflammatory and/or paraprobiotics compounds that, in combination with healthy diet and physical activity could provide a possible further strategy to ameliorate emotional–behavioral symptoms in children with ADHD.
Author Response
We thank the reviewer for the valuable comments. We have done out best to address all of the reviewers’ comments. Please see the attachment.

Round 2
Reviewer 1 Report
The authors have addressed the majority of my comments and the manuscript is stronger.
Unanswered comments remain:
- the sharing of code (even SPSS, stata, JAMOVI allow to share the code used for data analysis)
- the internal consistency of CBCL scales in the present study (and not in the validation study). I suspect it to be low.
- adjustment of p-values for multiple testing (if authors do not want to adjust their p-value, they can formulate their study as exploratory)
Reviewer 2 Report
The authors have satisfactorily responded to all my questions and made the necessary changes to the manuscript.
Author Response
We thank the reviewer for the kindly help.